# Combined Cardiac and Respiratory Monitoring from a Single Signal: A Case Study Employing the Fantasia Database

**DOI:** 10.3390/s23177401

**Published:** 2023-08-25

**Authors:** Benjamin M. Brandwood, Ganesh R. Naik, Upul Gunawardana, Gaetano D. Gargiulo

**Affiliations:** 1School of Engineering, Design and Built Environment, Western Sydney University, Penrith, NSW 2751, Australia; u.gunawardana@westernsydney.edu.au; 2Adelaide Institute for Sleep Health, College of Medicine and Public Health, Flinders University, Adelaide, SA 5042, Australia; ganesh.naik@flinders.edu.au; 3The MARCS Institute, Westmead, NSW 2145, Australia; 4Translational Research Health Institute, Westmead, NSW 2145, Australia; 5The Ingam Institute for Medical Research, Liverpool, NSW 2170, Australia

**Keywords:** respiration, cardiology, electrocardiogram, plethysmography, signal processing, fantasia, MATLAB

## Abstract

This study proposes a novel method for obtaining the electrocardiogram (ECG) derived respiration (EDR) from a single lead ECG and respiration-derived cardiogram (RDC) from a respiratory stretch sensor. The research aims to reconstruct the respiration waveform, determine the respiration rate from ECG QRS heartbeat complexes data, locate heartbeats, and calculate a heart rate (HR) using the respiration signal. The accuracy of both methods will be evaluated by comparing located QRS complexes and inspiration maxima to reference positions. The findings of this study will ultimately contribute to the development of new, more accurate, and efficient methods for identifying heartbeats in respiratory signals, leading to better diagnosis and management of cardiovascular diseases, particularly during sleep where respiration monitoring is paramount to detect apnoea and other respiratory dysfunctions linked to a decreased life quality and known cause of cardiovascular diseases. Additionally, this work could potentially assist in determining the feasibility of using simple, no-contact wearable devices for obtaining simultaneous cardiology and respiratory data from a single device.

## 1. Introduction

Biomedical data such as electrocardiograms (ECGs) and respiration rate (RR) measurements are routinely collected in hospitals worldwide for health assessment purposes. ECGs assist in diagnosing and managing cardiovascular diseases, while respiration rates are typically monitored as part of patients’ critical vital signs [1]. The demand for these medical measurements is on the rise, with cardiovascular diseases being the most common non-communicable diseases globally, responsible for 17.9 million deaths in 2019, while chronic respiratory disease was the third leading cause of death, with 4.1 million deaths in the same year [2].

Monitoring patients’ vital signs in hospitals is essential to evaluate their conditions and can help prevent health deterioration via timely intervention. However, nursing staff are often busy, so monitoring patients in general hospital wards is often incomplete with infrequent observations [3]. The “gold standard” measurement of respiration rates outside of critical care depends on medical staff counting chest undulations over a minute. However, this can be inaccurate primarily due to staff counting for a fraction of this time and extrapolating because of high workloads [4]. Further, to contend with clinical uncertainty, continuous respiration rate measurements instead of single-threshold values need to be collected [5].

Continuous extraction of RR from an ECG can be a more accurate and efficient method of determining respiration rates [6]. Respiratory information, while not immediately present as a waveform, is present as information contained in the ECG by three main mechanisms: respiratory sinus arrhythmia, changes in the distance of the electrodes from the heart during the respiration cycle, and changes to thoracic impedance [7]. Measuring ECG-derived respiration (EDR) using a single ECG lead can be done by R peak amplitude modulation, R-R peak frequency modulation measuring sinus arrhythmia, baseline wander, and bandwidth measurement using ECG and seismocardiograms (SCG) [8], among other methods.

On the other hand, cardiac oscillations are evident in many respiratory signals. Observations made during experiments using a simple, low-cost, contactless resistive chest band to monitor respiratory effort and measure tidal and cardiac stroke volume can show evidence of the heartbeat signal in calibrated volume data [9]. Cardiogenic oscillations reported in signals obtained using respiratory inductive plethysmography are visible during a held breath. The heartbeat signals in the respiratory data are synchronized with given ECG signals [10].

Meanwhile, there is a trend toward using wearable devices to gather patient telemetry within hospitals and, most importantly, outside when the patient returns home [11,12]. To this extent, combining cardiac and respiratory monitoring from a single device that collects respiration signals or cardiology signals and extracts the missing signal from the available one would be more efficient. Similarly, with the increase in wearable devices such as finger rings [13] and multisensory wearable devices [14], there will come an increased need to develop algorithms that extract accurate cardiologic and respiratory information whether that information has been obtained using ECG or some form of plethysmography, for the diagnosis and management of many health conditions.

Given that no previous work has attempted to locate heartbeat peak locations over long periods using a publicly available database, this study attempts to fill the gap in the literature. It proposes an initial process for identifying heartbeats in respiratory signals when the heartbeat signal is close to the noise floor. This work offers insights into the strengths and limitations of the proposed technique and presents ideas for improved algorithms that could increase the percentage of true positive heartbeat detection.

The findings of this study could contribute to the development of new, more accurate, and efficient methods for identifying heartbeats in respiratory signals, leading to better diagnosis and management of cardiovascular diseases, particularly during sleep where respiration monitoring is paramount to detect apnoea and other respiratory dysfunctions linked to a decreased life quality and known cause of cardiovascular diseases. Additionally, this work could potentially assist in determining the feasibility of using simple, no-contact wearable devices for obtaining simultaneous cardiology and respiratory data from a single device.

## 2. Related Work

In the current literature, numerous works addressed deriving respiratory information from ECG signals. Varon et al. [15] alone evaluate ten methods for recovering respiration from a single lead ECG and cite 12 articles using different methods for deriving respiration from ECG signals. A vast number of ways are discussed for the identification of QRS waveforms in noisy or otherwise compromised databases for the identification of many cardiovascular diseases, including, for example, abnormal heartbeat, myocardial infarction [16], arrhythmia [17], and COPD [18].

However, the literature on locating heartbeats from respiratory data is scant. Methods of heartbeat identification that have been used mainly belong to two categories: visual inspection [9,10,19,20] and empirical mode decomposition (EMD) [21,22].

### 2.1. Heartbeat Identification

Recent research discusses resistive and inductive methods for obtaining respiratory and cardiac tidal volume measurements from chest bands, where the presence of the heartbeat is shown by visual inspection between breaths [9,10,19]. Hornero et al. [20] discuss electrical impedance plethysmography for breathing, pressure registration, and cardiac activity from piezoelectric sensors in prosthetic limbs of 5 lower leg amputees.

Other studies [21,22] have used EMD to isolate the heartbeats obtained from various methods. Still, despite attempts by Puizzi et al. [21] to incorporate principal component analysis, it was reported that employing the EMD technique automatically and straightforwardly is not easy because it requires individual detection and a proper combination of the IMFs associated with heart activity. To compensate, Fontecave-Jallon et al. [23], Piuzzi et al. [21], and Kroshel [24] utilized signal processing methods such as bandpass filters and Fourier transforms to highlight the heartbeats present in non-ECG signals.

### 2.2. Derived Respiration

Respiration rate (RR) and ECG-derived respiration (EDR) are well-investigated areas in the literature. Due to this, finding comparative studies that combine many studies and compare the efficiency of many methods is not difficult. One study assessed over 300 algorithms for RR estimation [25], while another paper reviewed 12 studies that used alternate methods of acquiring EDR [15]. Four other studies examined more than three methods using the Fantasia database as a reference [26,27,28,29].

To extract the RR from a photoplethysmography (PPG), over 100 algorithms have been proposed [25]. Respiration rates can be derived using different approaches. One way is monitoring variations in the amplitude or frequency of R- and S-waves. Another method involves tracking bandwidth modulation or the difference in R-S amplitude. Additional techniques include comparing the area of the QRS complexes or analyzing changes in the slopes and angles of the QRS complexes [15]. More advanced methods incorporate deducing information from the QRS complexes using principal component analysis techniques such as discrete wavelet transformation, empirical mode decomposition, variational mode extraction, and variable-frequency complex demodulation [15].

Sharma and Sharma [27] measure the RR using a novel method employing Hermite basis functions and compare their algorithm to seven other methods of RR derivation. In contrast, another study that determined RR using EDR techniques by Schmidt et al. [28] employed fourth-order central moments, and Dong et al. [26] recreated a respiration waveform using seven known methods to verify the effectiveness of their two-stage EDR algorithm using multiple interacting models smoother on the features derived from the first moment (mean frequency) of the power spectrum.

There exists a plethora of studies attempting to determine RR from EDR processes and several studies that recreate some form of respiration waveform using cubic spline interpolation; however, there has been no attempt to create a composite waveform using multiple QRS variation data points to assemble an averaged waveform. A description of the process to determine true and false positive and false negative designations of peak locations was also not found. In both respects of EDR and RDC, there were no available comparisons of calculated peaks with recorded peaks in the time domain using sensitivity and precision on a standardized database.

## 3. Method

The aims of this research are accomplished by developing algorithms in MATLAB environment using the Fantasia database [30]. The Fantasia database is a publicly accessible database with the ECG and respiration recordings of 40 participants who watched the movie Fantasia when the data was taken. The recordings were taken from a single lead (lead II), and respiration data were recorded from the thorax. There were 20 young and 20 elderly study participants. Ten young (21–34 yr) and ten elderly (68–81 yr) “rigorously screened healthy” study participants were initially recorded before 1996, with another 20 participants added after 2000. The recordings were made to observe healthy sinus rhythm cardiac interval dynamics according to age [30]. The recordings were made publicly available via the Physionet databank; however, no more is known about the study participants.

The Fantasia database is being used for this study as it contains a tolerable difficulty level in reconstructing the EDR, as the subjects are awake but supine—the sleeping and driving datasets investigated by Varon et al. [15] were found to be easier and more challenging to work with, respectively, making the Fantasia database an excellent proving ground for newly developed algorithms.

### 3.1. Heartrate Identification

Attempts to incorporate EMD in this work do not perform as well as the method adopted in this study. Pre- and post-processing are still required to increase the accuracy of the EMD process. A bandpass filter is required as a preparatory step to avoid signal distortion. Additionally, post-EMD signal manipulations were needed to locate heartbeats effectively. Excluding the EMD step at that juncture improves positive outcomes. Therefore, this paper’s proposed method of finding heartbeats from respiratory data does not use EMD. In summary, our method incorporates the following steps: Signal Preparation, Signal manipulation, and lastly, detection as outlined below.

#### 3.1.1. Signal Preparation

Identifying heartbeats in the respiration signal involves bandpass filters, filtering subtractions, and array operations. The process is visualized in the following flowchart, as Figure 1.

**The HR is estimated using the respiration rate as a first step to locate the heartbeat.** This is achieved using a band pass filter allowing only frequencies from 0.2 to 0.5 Hz from the participants respiratory data. The number of breaths is determined using an automated algorithm. The probable heartbeat-to-beat interval is found by dividing the average time of breaths by three. Using a constant ratio of three heartbeats per breath as a caveat introduces inaccuracies when searching for heartbeat locations where the ratio varies. Heart rates per breath in the Fantasia database are four to 1.8 heartbeats per located breath; the average is 3.06. Future algorithm improvements would be updating heart inter-beat intervals using detected heartbeats.

The original respiration signal is passed through a bandpass filter, removing all information below 1 Hz and above 9 Hz. This effectively eliminates powerline noise (50 or 60 Hz) and baseline wander (0–0.15 Hz), as well as respiration (0.15 to 0.8 Hz).

The number of data points determined from heart inter-beat interval time at the sample rate of 250 Hz is used to determine the frame length for a 3rd-order Savitzky–Golay filter. This filter effectively preserves high-frequency signal components and is popularly used for smoothing data. A 3rd order setting achieves the required results without adding unnecessary computational requirements. The output from this filter is subtracted from the band-passed signal and is presented in Figure 2. The upper panel shows the raw respiration signal and the lower panel shows the filtered heartbeat signal, respectively. This process sharpens the heartbeat R wave, elongating the peak without introducing other peaks at the height of respiration (which plagues the process if a longer frame length is used).

#### 3.1.2. Signal Manipulation

The resulting filtered signal can still contain significant noise, and the filtered heartbeats in Figure 2 can be difficult to distinguish, with equally energetic R and T peaks in many cases. Interference from the noise floor may include respiration harmonics, which cannot be removed without affecting the heartbeat visibility in the resulting signal. Apparent noise or lack of heartbeat peaks may consist of signal differentiation due to the physical morphology of the participant.

A process to increase the definition of the R waves is outlined next. First, a moving average filter with a frame length of 30 ms is used to find the average of the filtered signal, as shown in Figure 3a. The 30 ms time period is chosen as long as the T peak but longer than the R peak. When the averaged signal is subtracted from the filtered signal, the height of merged peaks is reduced, and the bulk of the oscillations in the elongated T wave are lowered, as shown in Figure 3b. The process distinguishes the desired R wave peak and removes low-frequency noise from the signal. Subsequently, the signal derived in Figure 3b is added to the mean of its absolute values in an array operation, moving the entire waveform into the positive domain. The resulting array is then squared and amplified to obtain the signal in Figure 3c, where the longer R peaks are easily distinguished from the surrounding data. This signal is passed through another bandpass filter, which passes frequencies from 0.5 Hz to 5 Hz, and all negative values are set to zero. This signal is used to locate the heartbeats using the MATLAB peak function algorithm. Finally, Figure 3d shows the found and annotated heartbeats with the participant’s ECG data for the same period.

#### 3.1.3. Heartbeat Detection

Peaks are located using the MATLAB built-in (findpeaks) function with the minimum peak distance setting of 65% of the expected heartbeat-to-beat distance calculated earlier and over 8% of the average height. These percentages allow the search to re-center on the following heartbeat peaks after a false identification. The array of heartbeat locations derived is then corrected for an average physiological delay of 120 ms and compared to the supplied annotations in MATLAB. This physiological delay agrees with Hornero et al. [21], who found a delay of about 150 ms between both signals. The located heartbeats are shown in Figure 3c as small circles. Additionally, shown as asterisks are the supplied annotations for the ECG signal in the Fantasia database.

To inspect the sensitivity and precision of calculated heartbeat designations, the arrays containing the time locations for detected heartbeats were compared with the reference heartbeat annotation locations. One comparison was made for true and false positives, which used the found locations array to compare placement within the given annotations. Another comparison used the given annotations to locate false negative placements by comparing the located arrays.

Any deviations between the found and the given annotations within 140 ms (35 data points at a sample rate of 250 Hz) are treated as true positives. False negatives are determined when a calculated notation did not occur within 140 ms of the supplied Fantasia annotations, as shown in Figure 4. Any other placement is defined as a false positive or a false negative. False negatives are also recorded if no peak was discovered between annotations at an offset of 140 ms before peaks. In contrast, an incorrect peak is designated if it is within a minimum distance of 0.32 s or 80 data points.

Incorrectly identified T waves are recorded within a maximum distance after a heartbeat annotation of 320 ms. This information may be helpful in future improvements in the algorithm for error correction to measure an increase in true positives and a reduction in false negatives.

The provided heartbeat annotations and the heartbeats located by the algorithm are analyzed using a best-fit comparison, with both array sets split into one-minute sections, taken in increasing steps of 10 s per sample. The annotations are also compared for each new minute section with and without performing ‘best-fit’ adjustments but otherwise as per the process outlined by Sharma and Sharma [27]. Agreement of the determined HR and the expected HR are compared using Pearson’s coefficient rho values (ρ_p_) with a *p*-value for the ρ_p_ correlation significance, a normalized root mean square error (NRMSE) of the average HR, as well as the mean absolute error (MAE), percentage error (PE) and concordance correlation (ρ*_c_*). The equations used are detailed below.

Normalized root mean square.
(1)$$\mathrm{NRMSE}=\frac{\sqrt{\frac{1}{n}\sum_{i=1}^{n}{\left(bpm_{ref}^{i}-bpm_{calc}^{i}\right)}^{2}}}{bpm_{ref}{}_{\left(\max\right)}-bpm_{ref}{}_{\left(\min\right)}}$$

Mean absolute error.
(2)$$\mathrm{MAE}=\frac{1}{n}\sum_{i=1}^{n}\left|bpm_{ref}^{i}-bpm_{calc}^{i}\right|$$

Percentage error.
(3)$$\mathrm{PE}=\frac{1}{n}\sum_{i=1}^{n}\frac{\left|bpm_{ref}^{i}-bpm_{calc}^{i}\right|}{bpm_{ref}^{i}}\times 100\%$$

Pearson’s coefficient.
(4)$$\rho_{p}=\frac{\sum_{i=1}^{n}(bpm_{ref}^{i}-\frac{1}{n}\sum_{i=1}^{n}bpm_{ref})(bpm_{calc}^{i}-\frac{1}{n}\sum_{i=1}^{n}bpm_{calc})}{\left|\sum_{i=1}^{n}{\left(bpm_{ref}^{i}-\frac{1}{n}\sum_{i=1}^{n}bpm_{ref}\right)}^{2}-{\left(bpm_{calc}^{i}-\frac{1}{n}\sum_{i=1}^{n}bpm_{calc}\right)}^{2}\right|}$$

Concordance correlation.
(5)$$\rho c_{\;}=\frac{2S_{ref.calc}}{S_{ref}^{2}+S_{calc}^{2}+{\left\{\left(\frac{1}{n}\sum_{i=1}^{n}bpm_{ref}^{i}\right)-\left(\frac{1}{n}\sum_{i=1}^{n}bpm_{calc}^{i}\right)\right\}}^{2}}$$
where, for ρ_c_
(6)$$s_{ref\cdot calc}=\frac{1}{n}\sum_{i=1}^{n}\left[\left(bpm_{ref}^{i}-\frac{1}{n}\sum_{i=1}^{n}bpm_{ref})(bpm_{calc}^{i}-\frac{1}{n}\sum_{i=1}^{n}bpm_{calc}\right)\right]$$
(7)$$S_{ref}^{2}=\frac{1}{n}\sum_{i=1}^{n}{\left(bpm_{ref}^{i}-\frac{1}{n}\sum_{i=1}^{n}bpm_{ref}\right)}^{2}$$
(8)$$S_{calc}^{2}=\frac{1}{n}\sum_{i=1}^{n}{\left(bpm_{calc}^{i}-\frac{1}{n}\sum_{i=1}^{n}bpm_{calc}\right)}^{2}$$

### 3.2. Respiration Derivation

The method used here to isolate respiration information from the successive QRS complexes largely follows accepted and well-researched processes. However, differences in peak position and shape in the processed data are used without reference to the original ECG, as this increases overall accuracy for respiration peak detection. The respiration information is accessed in three ways, and the data is used to reconstruct a composite waveform. The process used is shown in Figure 5 below.

Outlier data is removed by replacement with previous data, with a simple for-loop using settings determined using visual inspection. This is a necessary step due to the level of interference in some respiration datasets and one ECG dataset, which significantly affects the mean. This enables better peak detection, using minimum peak height and distance. The respiration data obtained from the plethysmograph is passed through a bandpass filter with settings of 0.15 Hz and 0.5 Hz, and the MATLAB built-in function is used with settings of 2.5% of maximum height for minimum peak prominence and a minimum peak distance of 250. This data is later used to compare the accuracy of the reconstructed waveform.

The ECG data is processed with a bandpass filter to exclude frequencies below one and above 25 Hz. The QRS is then squared once and averaged twice with a moving mean filter with a frame length of 75 data points. Then, MATLAB is used to locate peaks with a minimum peak distance of 160 data points and a minimum peak prominence of 0.5% of the maximum of the processed waveform. The resultant peak data are used to determine changes in height and distance, which reveals the amplitude modulation of the R peaks and the frequency modulation of the heart’s sinus rhythm. The Q waveform and the squared ECG signals are shown in Figure 6.

To locate the Q trough minima, the ECG is filtered with a bandpass filter to exclude frequencies below 0.1 Hz and above 25 Hz to include bandwidth information within the respiration frequencies. Then, the data is altered to leave only the negative (below the x-axis) data, which is subsequently inverted on the horizontal. This waveform is averaged three times using the moving mean filter with a frame length of 75 points. The MATLAB inbuilt function is again used but with a setting of 10% of the maximum height of the waveform and a minimum search distance of 160 data points.

In all three cases of QRS wave peak and Q wave trough peak detection, outliers are again removed in some cases with settings determined using visual inspection. Then, all variance beyond 300% of the mean of the absolute value is removed. The absolute mean determination reduces the mean peak height to a percentage value for all three datasets. The three waveforms are then added to each other, averaged, and passed through a moving mean filter with a frame length of 300. The result is then passed through a bandpass filter excluding frequencies less than 0.15 Hz and greater than 0.5 Hz.

The detected peaks are then compared to those detected in the original respiratory data to assess the accuracy of determined peaks. This is accomplished by obtaining the average difference between expected inspiration peaks and reference peaks for the entire recording. The average mean distribution is determined using allocating bins of 0.2 s, and the entire composite waveform shifts to a maximum of 1.4 s. The inspiration peak locations are again assessed, using 50% of one average heartbeat period for true positive designations. Any designations outside this timing are recorded as a false negative. Peaks within 50–100% of one heartbeat are also labeled inaccurate peaks. The output is shown in Figure 7.

The respiration rates were assessed using the process used by other studies [15,27,28,29], where one-minute segments at increasing segments of ten seconds are compared for MAE, PE, and concordance correlation. Pearson’s correlation coefficients and NRMSE were also calculated per the methods of Schmidt et al. [28]. In addition, the waveform correlation performed by Dong et al. [26] and Varon et al. [15] was accomplished using the “xcorr” MATLAB function, using both randomized ten-minute sections of data and also by comparison of the entire database in 32-s sections using the +/− three-second best-fit adjustment that was also used by Sharma and Sharma [27].

## 4. Results

### 4.1. Heartbeat Location

#### True and False Positives

For participants with typical PQRST (PQRST describes the normal sequence of waveforms within a single heartbeat. The P wave designates atrial depolarisation, the QRS shows the period of ventricular polarisation, and ventricular repolarization is represented by the T section of the waveform complex.) waveform ECG signals, where the heartbeat waveform is distinguishable from background noise. The process outlined in this study performed exceptionally well in cases where the estimated HR agreed with the actual HR. Table 1 shows the true positive, false positive, precision, sensitivity, percentages, percentage error, and other metrics for all participants in the Fantasia database.

As seen in Table 1, the algorithm’s results were >60% for true positive heartbeat location for 12 participants with typical ECG heartbeat waveforms and heartbeat-to-breath ratios. True positives averaged 52.36% for all participants, with the incorrect T peak being located instead, an average of 8.44% of the time. On average, false positives at 40.5% and negatives at 45.73% remain significant issues for noisy or atypical database signals and low reference HR.

### 4.2. Correlation and Error

The MAE, PE, and NRMSE values reflect the precision and sensitivity of the algorithm as compared to the reference annotations, at 7.24 bpm, 12%, and 0.15 for the entire database, respectively. Pearson’s coefficient suggests a fair correlation, at 0.56, with an associated *p*-value of 0.01. These measurements reflect the true precision and sensitivity of the results and have not been adjusted for best fit.

In situations where a false positive heartbeat was detected in place of a true positive one, the frequency of the expected heartrates was maintained, so the true positive as the false positive heartbeat detections contribute to the low MAE and PE, as well as the high Pearson’s coefficient, which is all calculated using heart rates. However, the more significant metric is the precision and sensitivity of heartbeat detection, which indicates considerable room for improvement in the algorithm.

As the process of assessing the algorithm using heart rates does not reveal the accuracy of located heartbeats, it does not correctly show the benefit of the algorithm for the detection of cardiovascular disease and conditions such as extra-systolic beats. For this reason, comparing precision and sensitivity is an essential step in evaluating the algorithm’s effectiveness.

### 4.3. Respiration Derivation

#### True and False Positives

Overall, as shown in Table 2, the algorithm located 80% of respiration peaks to within one-half of a heartbeat of the reference inspiration maxima, with a simple average phase difference adjustment performed for each participant’s data where required. The algorithm also located an average of 104% of respiration peaks, similar to the result for PE of 4.78% for the best fit per minute analysis shown in Table 3. The results for the older subjects were slightly better than those for the younger subjects regarding sensitivity and reduced somewhat for precision.

### 4.4. Correlation and Error

The algorithm performed well compared to other methods that use a single EDR method in terms of MAE and percentage error, with an MAE of 0.89 and 1.07 and PE of 4.78% and 6.60% for young and old subjects, respectively. These values match the accuracy level of the results obtained by Orphanidou et al. [29], who excluded all one-minute sections that exceeded two bpm mean absolute error. Pearson’s coefficient is relevant at >0.6 for both groups and NRMSE values of 0.21 across the entire database, as per Table 3.

The best-fit waveform correlation was better using the entire signal for each participant at 0.76, with randomly acquired ten-minute sections having average values of 0.69, within the ranges of other methods described by [26] of between 0.51 and 0.8, as seen in Table 4.

### 4.5. Suggested Improvements and Future Work

As the heartbeats are more strongly evident during periods of held breath, the heart rate could be calculated from the inter-beat interval at these locations and used to determine a more accurate heart rate for the succeeding respiration period. The heart rate defines a minimum distance for peak location search. This algorithm assumes three heartbeats per breathing cycle. However, where this rate is lower, there are higher rates of missed heartbeats.

Additionally, as per the method used by Pan and Tompkins [31], heartbeats could be identified more accurately by a recursive search at distances indicated by nearby heartbeat locations.

The authors of this work intend to use the developed algorithms for private databases of respiration and ECG signals recorded using improved versions of the resistive bands developed by Garguilo et al. [9].

## 5. Conclusions

The algorithm developed for RDC does not perform well at low reference heart rates when significant noise is present, as the noise is located where a heartbeat is wrongly expected. The low concordance correlation indicates a small range of heart rates and a significant correlation to predicted rates. However, the NRMSE value of 0.14 for the entire database is encouraging, indicating close agreement with expected heart rates.

Obtaining clearer datasets with a higher SNR for further study would be beneficial, as would better methods of determining projected heart rates using sections of data at respiration minima, which could improve heartbeat detections. However, this study has demonstrated the ability of the developed algorithm to locate heartbeats in respiratory data and compensate for undetectable heartbeats. The resulting waveform could derive heartbeat signals from simple, low-cost wearable devices. The proposed improvements and suggestions for better data acquisition methods could lead to more accurate heartbeat identification and improved outcomes for diagnosing and managing cardiovascular disease. Overall, the foundation for further research has been made into identifying heartbeats in respiratory signals.

The algorithms described herein are intended to investigate recorded signals acquired using the wearable electroresistive band developed by [9,19], where heartbeats have been visually confirmed in the respiration data. It is hoped that the methods could also be used to investigate SCG signals, and perhaps also for other respiration or heartbeat signals, with some variation in process as required.

The EDR algorithm shows improved MAE and PE values than other methods, at the expense of slightly lower but still significant correlation coefficients. The algorithm has demonstrated comparable performance with different single-method respiration waveforms and derived respiration rates.

It has been shown that it is possible to retrieve a respiratory rate effective to within 4.7% of the actual RR of young study participants. The composite waveform accurately detects inspiration peaks, averaging 4.24% of the exact number of breaths at high precision.

Using both methods, the study has shown that obtaining heart and respiration rates is possible using a single signal. The accuracy of the heart rates may depend on eliminating noise from the respiration signal during retrieval, such as by using resistive chest bands.

## Figures and Tables

**Figure 1 sensors-23-07401-f001:**
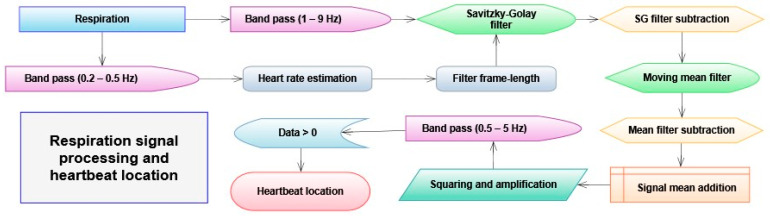
The signal processing method is used to locate the heartbeats in the respiration signal.

**Figure 2 sensors-23-07401-f002:**
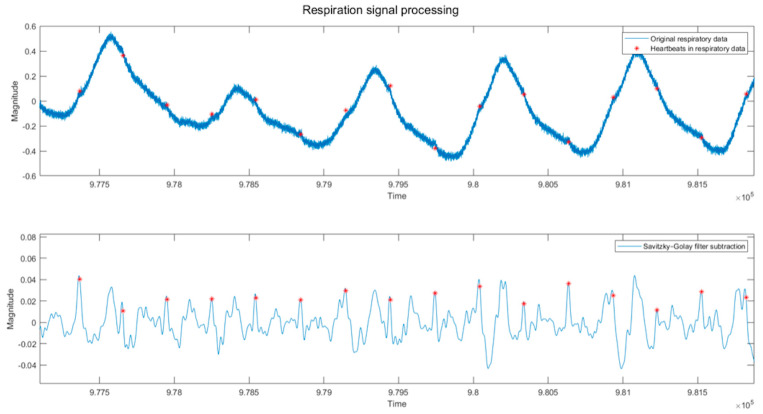
Filtered respiration is a cardiological signal. The heartbeats can just be observed in the original waveform between breaths indicated by vertical lines in the first panel. The filtered waveforms in the second panel are more precise but still contain noise artifacts.

**Figure 3 sensors-23-07401-f003:**
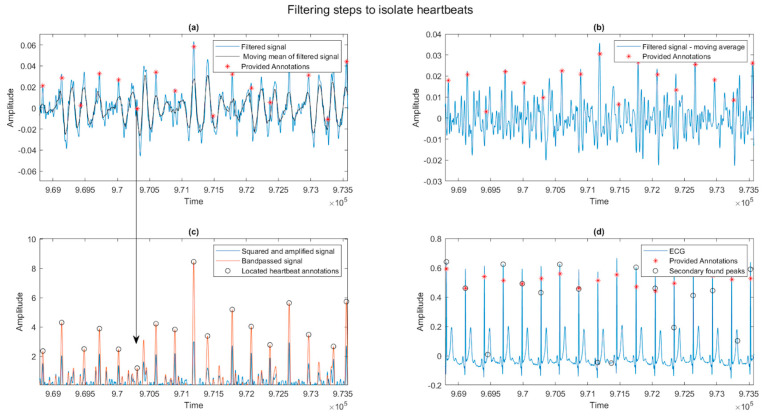
Sections of signal from participant 6 data show the filtering steps used to isolate the heartbeats for detection. The heartbeat indicated by an orange circle is no longer occluded, and the noise shown in blue has been significantly reduced. (**a**) shows the filtered signal, with the supplied annotations revealing correct heartbeat locations. Additionally, shown in black is the result of a moving mean filter 30 ms wide. (**b**) shows the result of subtracting the moving mean from the filtered signal. (**c**) shows the result in blue by adding the absolute mean of the signal in (**b**) and squaring the result. This signal, bandpass filtered to remove high-frequency noise, is shown in orange. Detected peaks are indicated with black circles. (**d**) shows the supplied ECG signal, supplied annotations in red asterixis, and the discovered annotations from the signal in (**c**) as black circles.

**Figure 4 sensors-23-07401-f004:**
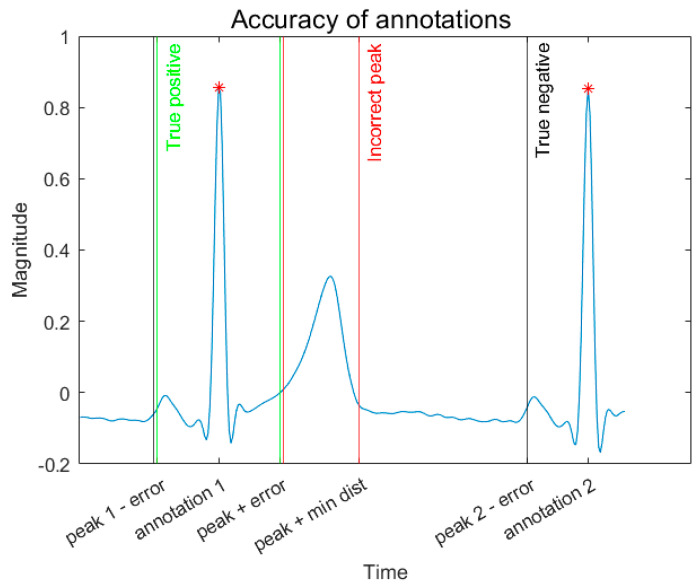
The determined areas for true and false positive results and false negative designations for heartbeats were discovered using the algorithm. The error used was 140 ms or 35 data points at a 250 Hz sample rate. The blue line is the ECG QRS waveforms showing two R peaks. The red asterixis show the reference annotation positions.

**Figure 5 sensors-23-07401-f005:**
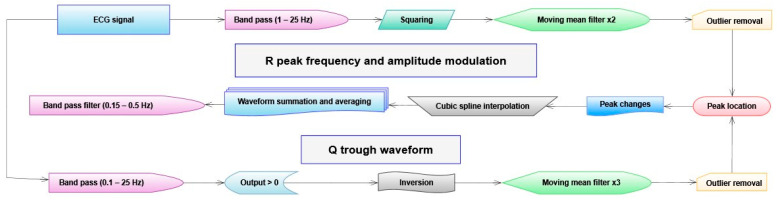
Shows the process used to determine a composite wave from the ECG data. The Q trough minima are isolated using a different process to the R peak amplitude and frequency modulation.

**Figure 6 sensors-23-07401-f006:**
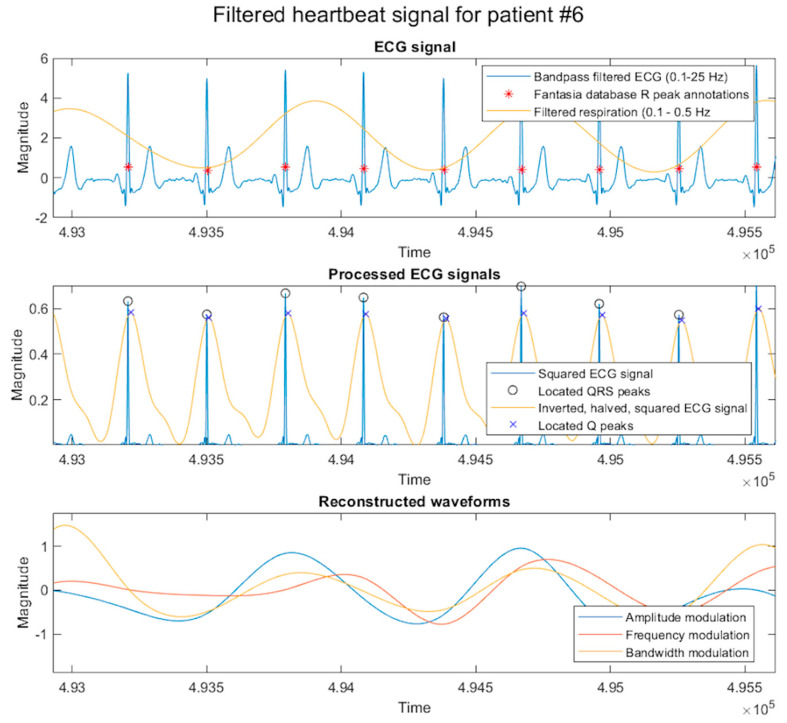
The filtered ECG and respiration signals. The central panel shows the squared electrocardiogram (ECG) waveforms, where the positive data is removed, and the signal is inverted on the x-axis before squaring. The third panel shows the cubic-spline reconstructed waveforms of the variance between QRS amplitude and frequency modulation. Additionally, shown is the Q trough variance obtained from the inverted, halved, and squared signals.

**Figure 7 sensors-23-07401-f007:**
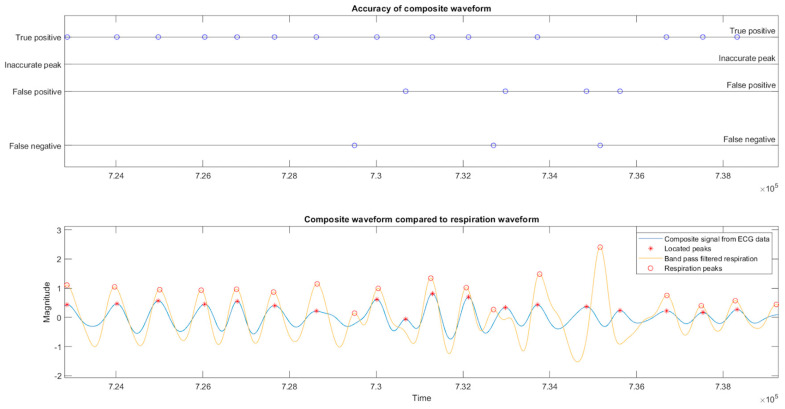
The reconstructed respiration waveform overlayed on the bandpass-filtered respiration waveform. On visual inspection, both the timing and amplitude are in close agreement. The blue circles indicate the true positive, false positive, true negative or inaccurate peak designation for each located peak indicated with red circles or missed peak as compared to the reference locations shown by red asterisk.

**Table 1 sensors-23-07401-t001:** The Pearson’s coefficient (ρ_p_), Concordance correlation (ρ_c_), mean absolute error (MAE), percentage error (PE), normalized root mean square error (NRMSE), and location data for the respiration-derived cardiology (RDC). The ‘best fit’ method yielded worse results than the unadjusted per-minute evaluations.

Results of Heartbeat Extraction	Young Subjects		Older Subjects		Total Averages
	Averages	‘Best Fit’	Averages	‘Best Fit’	
MAE (bpm)	5.84	6.35	8.64	9.94	7.24
PE	10.51	10.72	13.53	18.6	12.02
ρ*_c_*	0.34	0	0.25	0.02	0.3
ρ_p_	0.59	0.01	0.53	0.02	0.56
*p* value	0.00	0.24	0.01	0.27	0.01
NRMSE	0.13	0.71	0.16	0.95	0.15
True positive (%)	55.6	n/a	49.1	n/a	52.35
False positive (%)	40.2	n/a	40.8	n/a	40.50
True negative (%)	41.8	n/a	49.7	n/a	45.75
Precision (%)	59.8	n/a	55.4	n/a	57.60
Sensitivity (%)	57.2	n/a	49.8	n/a	53.50
Annotations (%)	98.6	n/a	91.4	n/a	95.00

**Table 2 sensors-23-07401-t002:** True positive, false positive, false negative, precision, and sensitivity for located respiration peaks were assessed using reference peaks for the Fantasia database.

Results of Respiration Detection	Averages		
	Young	Old	Total
True positive %	79.90	78.66	79.28
False positive %	23.44	28.86	26.15
False negative %	7.28	6.20	6.74
Precision %	77.63	75.28	76.46
Sensitivity %	91.11	92.29	91.70
Annotations %	103.44	105.04	104.24

**Table 3 sensors-23-07401-t003:** The waveform correlation for EDR utilising 12 methods across three studies, [1] Dong et al. used a random ten-minute section of data for each participant, assessing 32-s increments to calculate waveform correlation, [2] removed all one-minute sections with MAE > 2.0, [3] used ten minutes of data to calculate RR correlation, [3] adjusted +/− 3 s for best fit with every minute, with increments of 10 s each iteration, [4] use +/− 3 s for wave morphology correlation).

Waveform Correlation						
[1]	Method	QR	RR	RR + IMM	QRSW	QRS
	Waveform correlation	0.76	0.73	0.78	0.51	0.58
	Method	RS	R angle	QRSA	FMS	FMS + IMM
	Waveform correlation	0.54	0.56	0.73	0.77	0.8
Proposed method	Method	Composite				
	Waveform correlation					
	Total	0.76				
	Randomised 10 min	0.69				
[4]	Method	Normal				
	Waveform correlation	0.71				

**Table 4 sensors-23-07401-t004:** The Pearson’s coefficient (ρ_p_), Concordance correlation (ρ_c_), mean absolute error (MAE), percentage error (PE), and normalized root mean square error (NRMSE) statistics for the derived respiration rate using the Fantasia database for 15 methods across four studies, [1] used a random ten-minute section of data for each participant, assessing 32-s increments to calculate waveform correlation, [2] removed all one-minute sections with MAE > 2.0, [5] used ten minutes of data to calculate RR correlation, [3] adjusted +/−3 s for best fit with every minute, with increments of 10 s each iteration, [4] use +/−3 s for wave morphology correlation).

Correlation and Accuracy												
Algorithms	Younger Subjects						Older Subjects					
Composite Waveform (Proposed Algorithm)	MAE	PE %	ρ*_c_*	ρ_p_	*p* Value	NRMSE	MAE	PE %	ρ*_c_*	ρ_p_	*p* Value	NRMSE
‘Best fit per minute’	0.89	4.78	0.55	0.61	0.00	0.20	1.07	6.60	0.56	0.62	0.00	0.22
Average phase shift	1.33	7.79	0.51	0.58	0.00	0.10	1.18	7.21	0.51	0.58	0.00	0.10
RHDSD [4]	1.37	10.8	0.71				1.01	7.1	0.81			
RH DE	1.38	10.9	0.7				1.02	7.1	0.81			
RPCA	1.55	12.2	0.65				1.14	7.9	0.78			
RRamp	1.47	11.5	0.68				1.1	7.5	0.79			
RRSA	1.26	9.2	0.79				1.81	11.9	0.5			
RUS	1.85	12.7	0.6				1.99	13.8	0.44			
RDS	1.56	12.2	0.66				1.84	12.8	0.61			
RANG	1.83	12.7	0.59				1.92	13.6	0.56			
Mmom [3]	2.2	13.49					1.48	8.04				
Mslope	2.97	18.6					2.37	14.51				
Mrsa	2.27	13.16					3.42	19.73				
RSA [2]	0.83	4.9					0.97	5.7				
RPA	1.31	7.8					0.98	5.7				
RSA/RPA	0.81	4.8					0.84	4.9				

## Data Availability

The data presented in this study are openly available in Harvard Dataverse at https://doi.org/10.7910/DVN/IUJCAE, reference number 10.7910/DVN/IUJCAE.

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
