# Peer review of "Combined Cardiac and Respiratory Monitoring from a Single Signal: A Case Study Employing the Fantasia Database"

_sensors, 2023, doi:10.3390/s23177401_

Round 1

Reviewer 1 Report

The paper presents a method that can retrieve heartbeat and respiratory waveform from a single respiration signal. The topic is quite interesting and could be beneficial for further research and for reality. However, I am concerned that the algorithm does not give good accuracy in heartbeat detection nor detailed discussion of what need to be improved in the algorithm, especially when the authors claim that "the Fantasia database an excellent proving ground for newly developed algorithms". So, please revise the paper to address these in your major revision. More comments could be found in the attached file, highlighted in yellow.

I just have some minor comments in the attached file.

Reviewer 2 Report

The paper presented by Brentwood et al., describes addressing a challenging problem of combined cardiac and respiratory monitoring using a publicly available database to develop a computational approach of calculating both heart rate and respiratory rate. This is a reasonably well executed study, the conclusions are supported by the data presented in the paper. I have only minor comments:

11.      Figure 1 and 2 require units for the X axis.

22.       Lines 146-151. The description of the Fantasia database would benefit from more detail. The reference 30 provided by the authors provides only a very high level description of the database which provides no insights into the subjects and the data presented in this manuscript. Any clinical information concerning subjects who contributed the data for this specific study would be highly beneficial. What is the subject age? Any comorbidities, especially in elderly study participants? Devices and models used to collect the data used for the analysis presented in this paper? Timeframe when the data was actually collected?

33.        Lines 26-28. The authors make a claim in the abstract that this work could potentially assist in determining the feasibility of using simple noncontact wearable devices for obtaining simultaneous cardiology and respiratory to data from a single device. This topic is not discussed to any appreciable extent in the paper which is a shortcoming. The authors should elaborate more on this point: which specific contactless wearables can  leverage this work? If this is not feasible, the authors should remove this claim from the abstract.

Round 2

Reviewer 1 Report

Thank the authors for revising the manuscript taking into account my comments. Please find the attached file for my new comments highlighted in yellow.

I suggest the authors do a final review of your manuscript for this.

Author Response

Thank you again for taking the time to review our manuscript—apologies for the errors that occurred with the extensive changes that were made to the document.

Regarding the table of terms and abbreviations, The table now remains after accepting tracked changes.

Regarding the suggested improvements, it is difficult to know whether they would work as they have not been implemented. However, given our assumption of a constant heartrate of three heartbeats per breath may introduce inaccuracies where the heartbeat exceeds this rate ( the minimum for the Fantasia database is 1.8 using the inspiration peak detection settings, as discussed in lines 186) form of adjustable heart rate determination may decrease missed heartbeats. The section has been edited for clarity. Future work has also been added to the section.
